# *Nocardia farcinica* Brain Abscess in a Multiple Myeloma Patient Treated with Proteasome Inhibitor: A Case Report and Review of the Literature

**DOI:** 10.3390/brainsci11091204

**Published:** 2021-09-13

**Authors:** Nengwen Xu, Linjie Li, Wen Lei, Wenbin Qian

**Affiliations:** 1Department of Hematology, College of Medicine, The Second Affiliated Hospital, Zhejiang University, Hangzhou 310009, China; b1918070@zju.edu.cn (N.X.); leiwen2017@zju.edu.cn (W.L.); 2Department of Hematology, College of Medicine, Lishui Hospital, Zhejiang University, Lishui 323000, China; lilinjie0394@zju.edu.cn

**Keywords:** multiple myeloma, proteasome inhibitor, *Nocardia farcinica*, brain abscess

## Abstract

Nocardia brain abscess is relatively rare, accounting for about 1–2% of all brain abscesses, and its mortality rate is three times higher than of other types of bacterial brain abscesses; thus, early diagnosis and treatment are essential. Nocardia brain abscess generally occurs in immunodeficient patients. We report a case of *Nocardia farcinica* brain abscess in a multiple myeloma patient treated with proteasome inhibitor (bortezomib and ixazomib), cyclophosphamide, and corticosteroid. The patient was treated with ceftriaxone and trimethoprim-sulfamethoxazole, together with drainage of the brain abscess. Regular brain MRI follow-ups showed that intracranial lesions were gradually absorbed and improved.

## 1. Introduction

Nocardiosis is a localized or disseminated infection caused by an aerobic actinomycete that is soil-borne with a worldwide distribution [1]. Every year about 500 to 1000 new cases of Nocardia infections are reported [2,3] and most patients were in immunodeficiency states (such as diabetes and acquired immunodeficiency syndrome [AIDS]). Delayed diagnosis may be responsible for treatment failure and poor prognosis, while the mortality rate can be estimated as up to 60% [4]. In the past fifteen years, proteasome inhibitor treatment has had a favorable effect on the long-term survival rate of patients with multiple myeloma [5]. These drugs inhibit the ubiquitin-proteasome pathway affecting cellular and humoral immunity [6]. Bortezomib, a proteasome inhibitor in combination chemotherapy regimens, has been reported to be associated with the reactivation of herpes zoster virus and infection by various bacterial pathogens [7,8]. There are very few case reports of *Nocardia farcinica* brain abscesses associated with proteasome inhibitors and no cases have been reported from China. Here, we report a case of *Nocardia farcinica* brain abscess in a multiple myeloma patient treated with proteasome inhibitor, cyclophosphamide, and corticosteroid.

## 2. Case Presentation

The patient was a 69-year-old woman from Qingyuan, Zhejiang. By occupation, she is a homemaker. She was admitted to the Department of Nephrology in the Lishui Municipal Central Hospital on July 2020 due to fatigue and increased creatinine for more than 8 months. She used to be healthy, except for hypothyroidism two years ago. She was diagnosed with intermediate-risk IgG-lambda light chain multiple myeloma (ISS III stage) based on the following tests (Table 1) and bilateral lower extremity muscle venous plexus thrombosis in August 2020. Whole-body PET/CT on 22 July 2020 showed no lytic lesions and brain abscess in the patient. The patient presented with anemia and her renal failure met treatment initiation criteria. Beginning in August 2020, she received induction chemotherapy with bortezomib, cyclophosphamide, and dexamethasone (BCD) for 5 cycles and achieved a partial response. Maintenance therapy consisted of ixazomib and dexamethasone (ID), with partial response as per the International Myeloma Working Group (IMWG) criteria. At the same time, she was given anti-venous thrombosis therapy with rivaroxaban tablets. After receiving maintenance therapy with ID for 3 months, she developed general twitching with left limb weakness and required hospitalization. Her enhanced brain magnetic resonance imaging (MRI) showed two rings on the right frontal lobe enhancement foci with large edema and adjacent meninges enhancement, considering the possibility of brain abscess; there were a few lacunar foci in the deep white matter area of the frontal lobe on both sides. On 12 January 2021, chest computed tomography (CT) showed bronchial lesions, multiple pulmonary micronodules, chronic lung inflammation and emphysema, but she had no symptoms such as fever, cough, and expectoration. On 15 January 2021, emergency surgical drainage of the intracranial abscess was performed under general anesthesia and the microbiologic examination of the intraoperative cultures was positive for *Nocardia farcinica*. These bacteria were partially acid-fast by modified Kinyoun’s stain (Figure 1). The gram stain showed gram-positive slender filamentous bacteria (Figure 2). She was given ceftriaxone sodium (Rocephin), specifically 2 g qd intravenously, and trimethoprim-sulfamethoxazole (TMP-SMX), specifically 0.96 g q8h orally, for 10 days. Fortunately, the general condition of the patient became better. The muscle strength of the left limb was improved and then the patient was discharged. After 6 weeks of initial treatment with ceftriaxone and TMP-SMX, the patient was treated with TMP-SMX as a single-agent maintenance treatment and follow-up MRI scans were performed every 4–5 weeks. Regular brain MRI follow-ups showed that the intracranial lesions were gradually absorbed and improved (Figure 3).

## 3. Discussion

The Nocardia species are gram-positive branching filamentous bacillus widely distributed in nature environments such as air, water, and soil [9]. There is currently over 80 Nocardia species, of which 13 species such as *Nocardia farcinica* can cause human infections. *Nocardia farcinica* is the most common strain that causes central nervous system infection [10]. Nocardiosis can cause widespread disseminated infection. It mainly forms a local infection through the respiratory tract, through surgical operation, or on traumatic skin wounds, and it is easy to spread from the initial infection site to the lung, brain, kidney, joints, eyes, and other secondary infection sites [11,12,13]. The majority of infections are caused through the respiratory tract and therefore pulmonary involvement is the most likely initial manifestation [2]. The clinical features of *Nocardia farcinica* infections have no significant specificity, which can easily lead to underdiagnosis or misdiagnosis.

Nocardia extrapulmonary infection can be spread through blood or necrotizing pneumonia in many parts of the human body. The brain is the common involved site and abscess formation is an important feature. The prognosis of these patients is poor and the mortality rate is up to 31% [14]. In this case, a 69-year-old female with a history of hypothyroidism received proteasome inhibitors, glucocorticoids, and immunosuppressant cyclophosphamide because of multiple myeloma. Due to renal failure, she was given a dosage reduction of Valacyclovir for antiviral prophylaxis. The patient received no prophylaxis with trimethoprim/sulfamethoxazole (TMP/SMZ) and Levofloxacin due to renal failure. She received intravenous immunoglobulin for preventing infections in each period of the induction treatment. She stayed at home as a homemaker during the maintenance therapy period. Then, she developed systemic convulsions and left limbs after five months, and finally was diagnosed with an *Nocardia farcinica* brain abscess based on imaging and etiological evidence. Whole-nody PET/CT on 22 July 2020 showed no brain abscess in the patient; therefore, we can conclude that the brain abscess was not present prior to the diagnosis of MM. Compared to the chest computed tomography on 22 July 2020, it is with no marked difference that the chest CT on 12 January 2021 also showed bronchial lesions, multiple pulmonary micronodules, chronic lung inflammation, and emphysema (Figure 4). However, the patient had no symptoms such as fever, cough, and expectoration. Therefore, the possibility is very small that the *Nocardia farcinica* brain abscess developed through the respiratory tract, but we also cannot entirely exclude the possibility of pulmonary lesions. *Nocardia farcinica* can be found in water, soil, and dust. Thus, we speculate that the immunosuppressed patient acquired this infection via inhalation of microorganisms in dust or air through open wounds such as chronic pulpitis. The treatment of *Nocardia farcinica* brain abscess should firstly involve removing the lesion or performing pus drainage [15]. Secondly, antibiotics based on compound sulfamethoxazole can improve the survival rate [16] and amikacin, amipenem, and linezolid are alternative choices. Considering that immunocompromised patients often have severe underlying diseases and respond poorly to treatments, the mortality rate is high.

Nearly 20% of intracranial nocardiosis derives from Nocardia pulmonary disease and the main clinical manifestation concerns increased intracranial pressure. The symptoms of *Nocardia farcinica* brain abscess are often less obvious than bacterial brain abscesses. Therefore, all patients with pulmonary nocardia disease should accept brain imaging [17]. In previous reports [18,19,20], four patients with multiple myeloma who received bortezomib and dexamethasone developed central nervous system symptoms (Table 2). Proteasome inhibitors, such as bortezomib and carfilzomib, can reduce the effects of T cells, B cells, NK cells, and dendritic cells on the host immune system [21]. Compared with high-dose dexamethasone [7], bortezomib is associated with an increased incidence of shingles reactivation. In addition, a variety of bacterial pathogen infections including Pseudomonas aeruginosa, Streptococcus, and Enterobacter have been reported to be related with bortezomib. In this case, the combination treatment of multiple myeloma with both proteasome inhibitor and glucocorticoids may increase the risk of nocardiosis.

The diagnosis of nocardiosis is based on the identification of the infection site. Gram staining and modified acid-fast staining are usually used for the preliminary identification of Nocardia [22]. Nocardia can grow on most non-selective culture media and takes at least 48 h to several weeks to grow significantly. Nocardia colonies have diverse morphologies [23]. Although DNA sequencing is currently the best tool for the identification of Nocardia species, the 16S RNA sequencing is a recommended and feasible method [24].

Most nocardia bacteria are sensitive to TMP-SMX and linezolid. Therefore, the empirical antibacterial treatment plan is based on the TMP-SMX and the antibiotic therapy was selected upon susceptibility tests. For pulmonary and central nervous system nocardiosis, the recommended regimen should include drugs with good permeability to lung tissue and the blood–brain barrier. These antibacterial drugs include TMP-SMX, ceftriaxone, meropenem, and fluoroquinolone. The duration of treatment for nocardia disease is mainly dependent on the severity of the infection and the immune status. It usually takes 6 to 12 months for treatment. Antibacterial treatment should be maintained at least for 6 weeks after the symptoms and signs disappear. For patients with migratory lesions or those in an immunosuppressed state, the treatment duration is generally extended for at least 12 months after stopping the immunosuppressant to minimize the risk of recurrence [16]. In this case, we performed simple drainage and added ceftriaxone to the TMP-SMX. We also suspended maintenance chemotherapy for 3 months. In our patient, the case is similar to previously published cases in that she received proteasome inhibitor and dexamethasone when she acquired the *Nocardia farcinica* brain abscess (Table 1). However, due to the fact that our patient experienced impaired renal function, TMP-SMX dosage adjustment was necessary. Renal function was measured every week. We thought that surgical intervention, early identification for the pathogen, and an empirical antibacterial treatment plan responsible for the favorable evolution of this immunosuppressed patient would be ideal. As a result, the patient improved after surgical intervention and antibiotics.

## 4. Conclusions

We report a case of *Nocardia farcinica* brain abscess in a multiple myeloma patient treated with proteasome inhibitor-based combination treatment. With this case, we hope to increase awareness among hemotologists about this rare infection in immunocompetent hosts. Early identification for the pathogen and appropriate identification of the specific nocardial species are crucial to start appropriate antibiotic therapy to improve patient survival.

## Figures and Tables

**Figure 1 brainsci-11-01204-f001:**
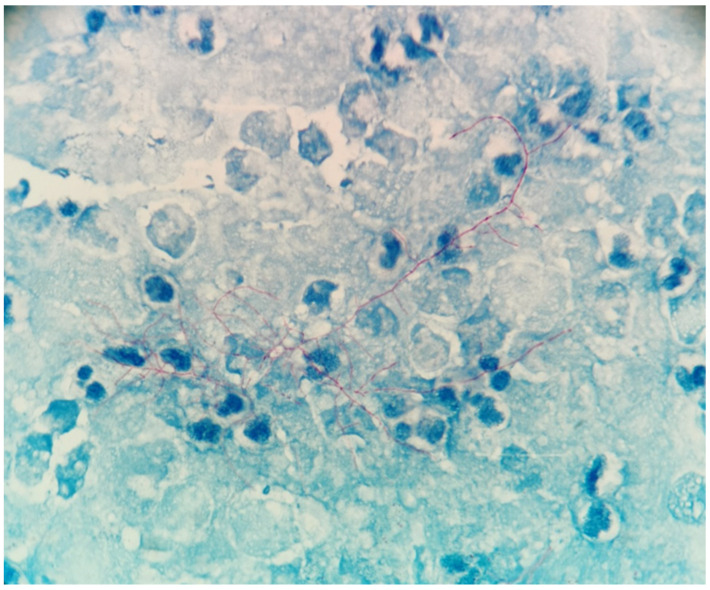
Partial acid fastness of the bacteria was demonstrated using modified Kinyoun’s procedure (×1000).

**Figure 2 brainsci-11-01204-f002:**
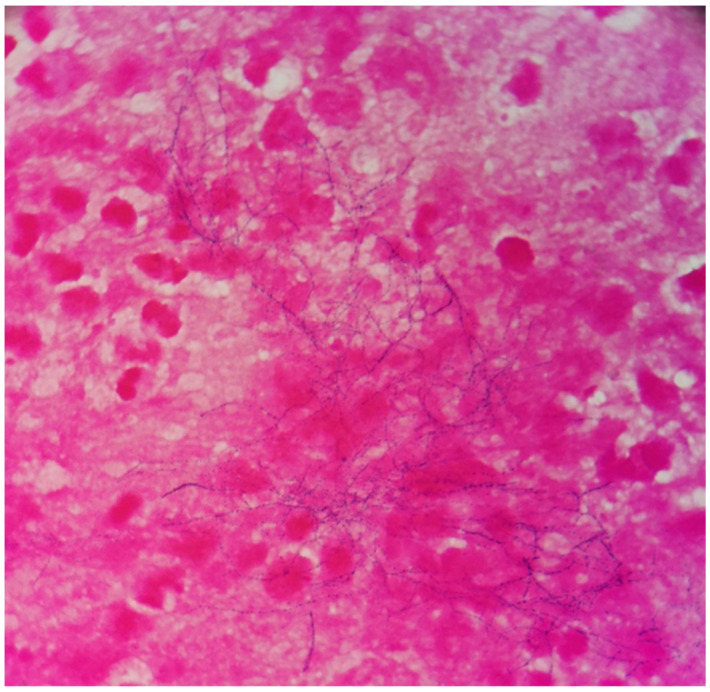
The gram stain revealed gram-positive filamentous and beaded bacteria (×1000).

**Figure 3 brainsci-11-01204-f003:**
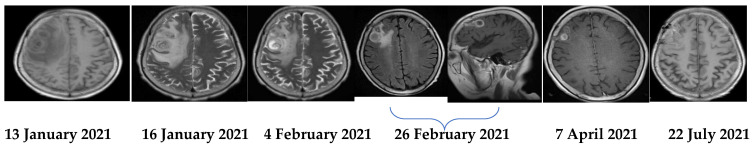
Serial MRIs during the treatment course.

**Figure 4 brainsci-11-01204-f004:**
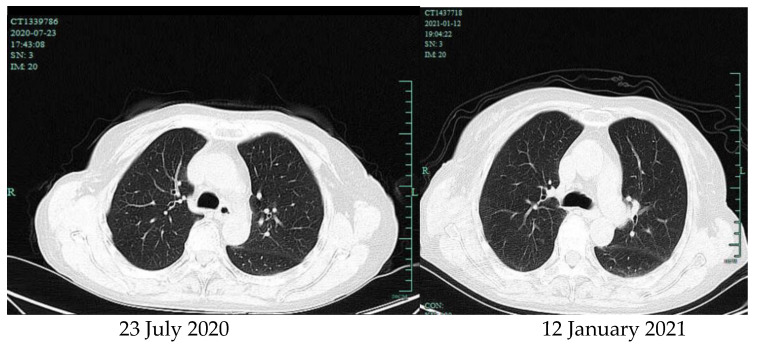
Chest computed tomography.

**Table 1 brainsci-11-01204-t001:** Initial laboratory workup, including reference ranges.

Test/Examination Name	Test Results (Reference Range)
White blood cell count	6.3 (3.5–9.5 ×10^9^/L)
Hemoglobin	86 (115–150 g/L)
Platelet count	192 (125–350 ×10^9^/L)
Serum monoclonal immunoglobulin oncentration	IgG23.5 (7–15 g/L)
	IgA0.37 (0.7–4.5 g/L)
	IgM0.22 (0.5–3.0 g/L)
	Kappa light chain 0.78 (1.7–3.7 g/L)
	Lambda light chain 5.47 (0.9–2.1 g/L)
Serum immunofixation electrophoresis (SIFE)	IgG lambda (Negative)
Plasma cells in the bone marrow	11 (0–2.1%)
Serum creatinine	243 (41–61 μmol/L)
eGFR	16.6 (>90 mL/min)
Renalbiopsy	Lambda light chain renal amyloidosis
Cytogenetic abnormalities	1q21 amplification
Whole-body PET/CT	No lytic lesions, no brain abscess
β2-microglobulin	9.2 (1.0–3.0 mg/L)

**Table 2 brainsci-11-01204-t002:** Clinical features of previously published cases of central nervous system nocardiosis in MM patients.

Case Number	Age/Sex	Underlying Disease	Chemotherapy Regimen	Clinical Presentation	Nocardia Spp.	Treatment Regimen	Outcomes	Reference
1	61/F	MM	Cyclophosphamide, bortezomib, and dexamethasone	Seizure	Nocardia cyriacigeorgica	Meropenem, then oral Amoxicillin clavulanic acid	Survived	Pamukçuoğlu et al.
2	60/F	MM	Cyclophosphamide, bortezomib, and dexamethasone	Dysarthria and gait disturbance	Nocardia cyriacigeorgica	Imipenem/cilastatin then TMP-SMX	Survived	Pamukçuoğlu et al.
3	69/M	MM	Lenalidomide, bortezomib, and dexamethasone	Fever, dyspnea, alteration of consciousness, and right-side weakness	Nocardia farcinica	TMP/SMX and Moxifloxacin	Survived	S. Chansirikarnjana et al.
4	68/F	MM	Melphalan, bortezomib, thalidomide, adriamycin, cyclophosphamide, cisplatin, and etoposide	Right-sided facial palsy with right-sided hemiplegia	Nocardia farcinica	TMP/SMX, linezolid, and imipenem, in addition to surgical debridement, and then TMP/SMX and tedizolid	Survived	A. Matin et al.
5	69/F	MM	Bortezomib, Ixazomib, Dexamethasone, and Cyclophosphamide	Convulsion of the limbs with left-sided hemiplegia	Nocardia farcinica	TMP/SMX and ceftriaxone, and then TMP/SMX and Moxifloxacin	Survived	Our case

## Data Availability

The data presented in this report are available from the first author upon reasonable request.

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
