# Peer review of "Nocardia farcinica Brain Abscess in a Multiple Myeloma Patient Treated with Proteasome Inhibitor: A Case Report and Review of the Literature"

_brainsci, 2021, doi:10.3390/brainsci11091204_

Round 1

Reviewer 1 Report

This is an interesting and overall well-written case presentation regarding the development of a brain abcess with Nocardia farcinica in a multiple myeloma patient. Some suggestions for revision are listed below:

  1. Please present clearly all the diagnositic criteria/lab values/data that justified the diagnosis of multiple myeloma: complete blood count, serum protein levels, immune electrophoresis, bone marrow examination etc.
  2. Please present the disease activity criteria that justified treatment initiation in this patient.
  3. Please improve the discussions. How do you explain the development of a brain abscess in this patient as N.farcinia infection affects the respiratory system primarily and then can disseminate to the brain? How was the exclusion of pulmonary lesions that could have not been visible on the CT scan done? 
  4. How do the authors explain the favorable evolution of this immunosuppressed patient? 

Author Response

  1. Please present clearly all the diagnositic criteria/lab values/data that justified the diagnosis of multiple myeloma: complete blood count, serum protein levels, immune electrophoresis, bone marrow examination etc.

        Response: Thank for your valuable advice.We have added the presentation of the MM diagnosis in the Table 1and treatment initiation criteria in the line 7, The first paragraph of case presentation.

    2.Please present the disease activity criteria that justified treatment initiation in this patient.

Response: Treatment initiation criteria of the patient is in the line 7, The first paragraph of case presentation.

  3.Please improve the discussions. How do you explain the development of a brain abscess in this patient as N.farcinia infection affects the respiratory system primarily and then can disseminate to the brain? How was the exclusion of pulmonary lesions that could have not been visible on the CT scan done? 

Response: The patient stayed at home as a homemaker during maintenance therapy period. Comparing with Chest Computed Tomography on July 22rd 2020,it is no marked dif-ference that Chest CT on January 12th, 2021 also showed bronchial lesions, multiple pulmonary micronodules,chronic lung inflammation and emphysema(Fig. 4).But the patient had no symptoms like fever, cough and expectoration . Therefore, Such possi-bility is very small that Nocardia farcinica brain abscess through the respiratory tract . but we also cannot entirely exclude the possibility of pulmonary lesions.Nocardia farcinica can be found in water, soil, and dust , So we speculate on the immunosuppressed pa-tient acquired this infection via inhalation of the microorganisms in dust or air through open wounds such as chronic pulpitis .

4.How do the authors explain the favorable evolution of this immunosuppressed patient? 

Response: We thonght that surgical intervention, early identification for the pathogen and em-pirical antibacterial treatment plan responsible for the favorable evolution of this im-munosuppressed patient.

Reviewer 2 Report

This is an interesting case report regarding the occurrence of a brain abscess caused by N. farcinica in a multiple myeloma patient. Some suggestions for revision are listed below:

  1. I believe Nocardia farcinica should be written in italics.
  2. The table should be moved to the discussion section.
  3. The authors need to present the local antibiotic and antiviral prophylaxis in MM. If the patient had received TMP-SMX as prophylaxis, would have this infection occurred?
  4. The case report needs to be improved in terms of the presentation of the MM diagnosis and treatment initiation criteria. This part of the case report is only briefly discussed. Maybe the patient had the abscess prior to the diagnosis of MM?
  5. The authors need to speculate on how the patient acquired this infection. What was her occupation? Was she at risk for acquiring this germ or did she acquire it only because she was immunosuppressed?
  6. The references need to be revised to match the style of the journal, namely American Chemical Society.  

Author Response

  1. I believe Nocardia farcinica should be written in italics.

         Response:According to the reviewer’s comment,we have correct it.

  1. The table should be moved to the discussion section.

         Response:According to the reviewer’s comment,we have correct it.

  1. The authors need to present the local antibiotic and antiviral prophylaxis in MM. If the patient had received TMP-SMX as prophylaxis, would have this infection occurred?

         Response: Because of renal failure,she was given dosage reduction of       Valacyclovir for antiviral prophylaxis.The patients received no prophylaxis with trimethoprim/sulfamethoxazole (TMP/SMZ) and Levofloxacin due to renal failure. She received intravenous immunoglobulin for preventing infections in each period of induction treatment. If the patient had received TMP-SMX as prophylaxis, it would reduce the risk of this infection.

  1. The case report needs to be improved in terms of the presentation of the MM diagnosis and treatment initiation criteria. This part of the case report is only briefly discussed. Maybe the patient had the abscess prior to the diagnosis of MM?

         Response:Thank for your valuable advice.We have added the presentation of the MM diagnosis in the Table 1 and treatment initiation criteria in the line 7, The first paragraph of case presentation.

  1. The authors need to speculate on how the patient acquired this infection. What was her occupation? Was she at risk for acquiring this germ or did she acquire it only because she was immunosuppressed?

         Response: The patient stayed at home as a homemaker during maintenance therapy period. Comparing with Chest Computed Tomography on July 22rd 2020,it is no marked difference that Chest CT on January 12th, 2021 also showed bronchial lesions, multiple pulmonary micronodules,chronic lung inflammation and emphysema(Fig. 4).But the patient had no symptoms like fever, cough and expectoration . Therefore, Such possibility is very small that Nocardia farcinica brain abscess through the respiratory tract . but we also cannot entirely exclude the possibility of pulmonary lesions.Nocardia farcinica can be found in water, soil, and dust , So we speculate on the immunosuppressed patient acquired this infection via inhalation of the microorganisms in dust or air through open wounds such as chronic pulpitis .

  1. The references need to be revised to match the style of the journal, namely American Chemical Society.  

         Response:According to the reviewer’s comment,we have correct it.

Round 2

Reviewer 2 Report

The paper has been improved. During copyediting/proofreading, I would suggest deleting the exact dates of when the patient attended your department to prevent a possible identification of the patient. Just say July instead of July 22nd. It's better to present data like this to ensure that the patient's confidentiality is not breached.